# Surface acoustic wave photonic devices in silicon on insulator

Dvir Munk[1,2,4], Moshe Katzman[1,2,4], Mirit Hen[1,2], Maayan Priel[1,2], Moshe Feldberg[2], Tali Sharabani[2,3], Shahar Levy[1,2], Arik Bergman [1,2] & Avi Zadok [1,2]

Opto-mechanical interactions in planar photonic integrated circuits draw great interest in basic research and applications. However, opto-mechanics is practically absent in the most technologically significant photonics platform: silicon on insulator. Previous demonstrations required the under-etching and suspension of silicon structures. Here we present surface acoustic wave-photonic devices in silicon on insulator, up to 8 GHz frequency. Surface waves are launched through absorption of modulated pump light in metallic gratings and thermo-elastic expansion. The surface waves are detected through photo-elastic modulation of an optical probe in standard race-track resonators. Devices do not involve piezo-electric actuation, suspension of waveguides or hybrid material integration. Wavelength conversion of incident microwave signals and acoustic true time delays up to 40 ns are demonstrated on-chip. Lastly, discrete-time microwave-photonic filters with up to six taps and 20 MHz-wide passbands are realized using acoustic delays. The concept is suitable for integrated microwave-photonics signal processing.

[1] Faculty of Engineering, Bar-Ilan University, 5290002 Ramat-Gan, Israel. [2] Institute for Nano-Technology and Advanced Materials, Bar-Ilan University, 5290002 Ramat-Gan, Israel. [3] Department of Chemistry, Bar-Ilan University, 5290002 Ramat-Gan, Israel. [4] These authors contributed equally: Dvir Munk, Moshe Katzman. Correspondence and requests for materials should be addressed to A.Z. (email: Avinoam.Zadok@biu.ac.il)

The research field of opto-mechanics addresses the interaction between light and elastic waves in a common medium[1–4]. The topic draws great interest in recent years, in basic research of light-matter interactions and quantum mechanics, development of quantum technologies, sensing and signal processing applications. Planar photonic integrated circuits provide an excellent platform for the study of opto-mechanics, with large freedom of design, small footprint, repeatable fabrication, and integration alongside electronic and optical functionalities. Optical waves and gigahertz-frequencies acoustic waves may be coupled effectively in integrated circuits, due to the similarity in their wavelengths (for a recent comprehensive review see ref. [1]). Opto-mechanical interactions are therefore suitable for the processing of signals at microwave frequencies[1,5,6]. Coupling between light and mechanical waves in planar integrated circuits was successfully demonstrated based on piezo-electric transduction in gallium-arsenide (GaAs) and aluminium-nitride (AlN)[7–14], and optical forces in diamond[15,16], silica[17,18], silicon-nitride (SiN) in silica[19] and chalcogenide glasses[5,20–24].

Silicon on insulator (SOI) is arguably the most significant material platform for photonics, due to the promise of monolithic integration alongside electronic circuits[25–27]. However, opto-mechanical interactions are yet to be introduced to standard SOI photonics. Silicon does not exhibit a piezo-electric effect. In addition, the silicon device layer in SOI does not effectively guide acoustic modes, which tend to leak towards the bulk[28]. Remarkable demonstrations of forward stimulated Brillouin scattering in silicon have been reported in recent years[29–32]. However, these required that the underlying oxide layer of SOI is etched away and that silicon waveguides and membranes remain suspended[29–32]. In one notable exception, mechanical motion of thin silicon fin waveguides has been excited by radiation pressure[33]. Alternatively, stimulated Brillouin scattering has been achieved in regions of chalcogenide glasses that were integrated as part of a passive SOI layout[22].

One of the challenges of integrated photonics is the relatively long delay of waveforms, (sometimes referred to as light storage), over tens of ns or longer[34]. Group delays serve as basic building blocks of discrete-time microwave-photonic filters[35] and beam steering modules[36,37]. Long delays of fast-moving light waves require waveguide paths that are many metres-long. Propagation delays over 100 ns were achieved in long silica or SiN in silica waveguides with ultra-low losses[19,38,39], however such optical path lengths cannot be realized in SOI. Waveform delays were also demonstrated in active recirculating loops that were switched on and off by semiconductor optical amplifiers[40]. Those devices required the hybrid integration of indium-phosphide (InP)-based active layers on top of silica.

The challenge of long time delays of incident analogue waveforms was already faced by electronic integrated circuits more than fifty years ago. A very successful solution path relies on the introduction of surface-acoustic waves (SAWs)[41–43]. Input radio-frequency (RF) waveforms are converted to slow-moving acoustic waves using an array of inter-digital electrodes on top of bonded piezo-electric materials, and recovered back to the electrical domain using a similar, second arrangement[41–43]. The slow speed of sound allows for the realization of delays over physical extents that are five orders of magnitude shorter than electrical paths. So-called SAW devices serve in the filtering, correlation, and analogue processing of RF signals. Advances in high-rate analogue-to-digital conversion and digital signal processing have partially pushed aside SAW-electronic devices. However, ultra-broadband optical signals cannot be digitized directly. Piezo-electric excitation of SAWs in GaAs was previously used in modulation and switching of multiple Mach-Zehnder interferometer waveguides[8–10].

In this work, we present SAW-photonic devices in standard SOI, which do not require piezo-electric actuation, the suspension of membranes, or the hybrid integration of additional materials. Incident RF waveforms at gigahertz frequencies are converted from the modulation of pump light to the form of surface waves, and then to the modulation of an optical probe that is propagating in a standard photonic circuit. The method of surface waves stimulation is generic and applicable to other substrates as well. Acoustic delays of tens of nanoseconds are achieved on-chip. The SAW-photonic delays are used in four-tap and six-tap discrete-time integrated microwave-photonic filters, with tens of megahertz-wide passbands. The complex weights of individual taps can be designed arbitrarily. The results represent the first introduction and application of opto-mechanics in standard silicon photonics.

## Results

**Principle of operation.** The principle of operation of SAW-photonic devices in SOI is illustrated in Fig. 1. A grating of thin gold stripes with period $\Lambda$ is deposited on an SOI substrate. The grating is illuminated by an optical pump beam, which is amplitude-modulated by a sine wave of radio-frequency $f$. Absorption of the modulated pump wave leads to periodic temperature changes in the metallic stripes[44–46]. Thermalization of the thin stripes occurs within few picoseconds[44–46]. The temperature perturbations are accompanied by periodic expansion and contraction of the metals, and the resulting strain pattern is transferred to the underlying silicon device layer[44–46]. Strain on the silicon surface is periodic in both space and time, with periods $\Lambda$ and $f$, respectively. When the product $\Lambda f$ matches the phase velocity of a surface-acoustic mode of the SOI layer stack, a

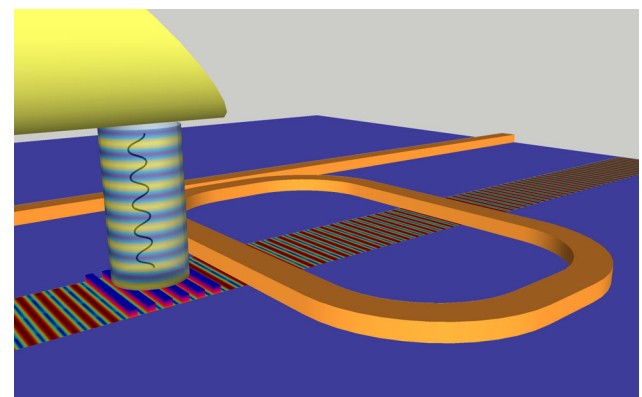

**Fig. 1** Surface-acoustic wave-photonic devices. Modulated pump light from the output facet of an optical fibre is incident upon a gold grating. The fibre facet is polished at an angle of 40°. The absorption of pump light leads to periodic heating and cooling of the grating stripes, which are accompanied with thermal expansion and contraction. The resulting strain pattern is transferred onto the underlying device layer of a silicon on insulator substrate. The strain pattern is periodic in time according to the modulation of the pump wave, and in space according to the grating pattern. When the temporal and spatial periods match those of a surface-acoustic mode of the silicon on insulator layer stack, a surface-acoustic wave is launched away from the grating region. A race-track resonator waveguide is defined in the silicon device layer, in proximity to the grating. The surface waves induce photo-elastic modulation to the effective index of the optical mode in the race-track waveguide. An optical probe wave, with a frequency that matches a slope of the resonator transfer function, is coupled into the waveguide. Photo-elastic perturbations due to surface-acoustic waves induce intensity modulation of the optical probe wave. The surface waves may cross the race-track waveguides in several locations, separated by comparatively long acoustic propagation delays

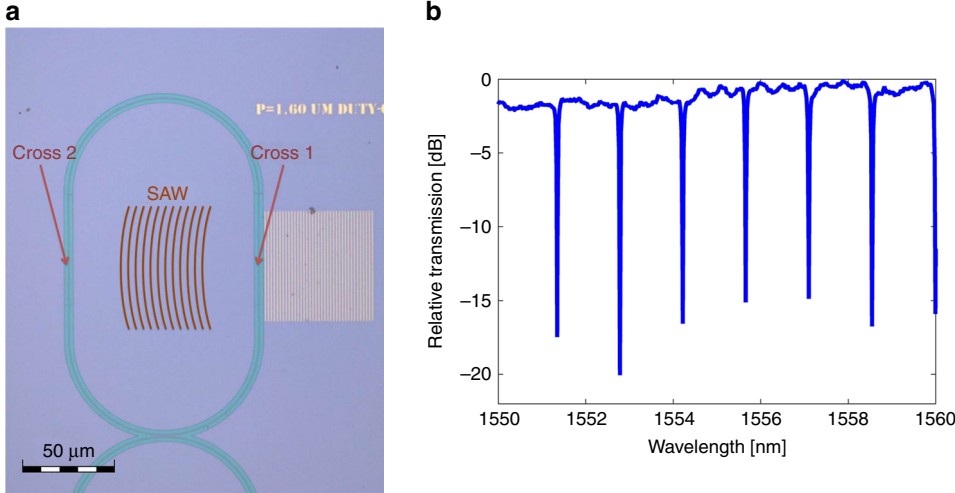

**Fig. 2** Optical characterization of devices under test. **a** Top-view image of a race-track resonator waveguide in silicon on insulator. A pattern of thin gold stripes with a period of 1.6 μm is defined near the resonator waveguide. The scale bar represents 50 μm. Surface-acoustic waves propagating away from the grating region are illustrated. The surface wave-front crosses two straight sections of the race-track waveguide. The acoustic propagation delay between the two events is 28.5 ns. **b** Measured normalised optical power transfer function of the race-track resonator. The loaded quality factor of the resonator is ~40,000. The extinction ratio reaches 20 dB. Source data are provided as a Source Data file

surface wave may be launched away from the illuminated grating region. The excitation of SAWs through the absorption of short pump pulses in structured metallic patterns has been used on top of bulk silicon wafers[44–46]. Herein, we introduce this technique to integrated photonics.

A standard race-track resonator optical waveguide is defined in the silicon device layer, in proximity to the metallic grating (see Fig. 1). The stimulated SAWs cross the waveguide path in one or more sections. The acoustic waves induce photo-elastic perturbations to the effective index of the optical mode in the resonator waveguide. An optical probe wave, at a frequency within a spectral slope of the resonator transfer function, is coupled into the waveguide. The index perturbation gives rise to intensity modulation of the output probe wave. The incident RF modulation of the optical pump is thereby wavelength-converted onto the probe, via stimulated SAWs.

The principle of operation is analogous to that of traditional SAW-electronic devices, with adaptations to SOI photonics: Piezo-electric transduction and detection are replaced by the absorption of one modulated optical carrier in metal, and the photo-elastic modulation of another in a standard waveguide, respectively. The stimulation of acoustic waves through a given grating strongly depends on the frequency $f$. Devices therefore function as integrated microwave-photonic bandpass filters[32,47,48]. Filtering may be enhanced further by multiple crossings of optical waveguide paths with different discrete acoustic delays, as discussed later.

**Experimental results**. Figure 2a shows a top-view microscope image of a device under test. A $60 \times 60$ μm² grating of gold stripes was patterned 4 μm away from one straight section of a race-track resonator. The grating stripes were defined in parallel with the < 110 > crystalline axis of the silicon device layer. The resonator comprised of ridge waveguides with partial etch depth of 70 nm and 700 nm width. The race-track waveguide was 480-μm-long. The fabrication process is detailed in the Methods. Light was coupled between standard single-mode fibres and the resonator bus waveguide using vertical grating couplers. Coupling losses were 10 dB per facet. Figure 2b shows an optical vector network analyser measurement of the normalized power transfer function

of the resonator. The full width at half maximum of the resonant transmission notches is 5 GHz, corresponding to a loaded $Q$ factor of about 40,000. The extinction ratio of the transfer function is ~20 dB.

SAWs were stimulated using an optical pump wave at 1540 nm wavelength (see setup illustration in Fig. 3). The pump wave was amplitude-modulated by a sine wave of variable radio-frequency $f$ from the output port of a vector network analyser. The pump was amplified by an erbium-doped fibre amplifier (EDFA) to an average optical power $P_{\text{pump}}$ between 50 and 500 mW. The end facet of the amplifier's output fibre was positioned above the gold grating. The vertical separation between the fibre facet and the device plane was adjusted so that the pump beam diameter matched the extent of the grating. An optical probe wave of 1532 nm wavelength and off-chip power of 100 mW was coupled into the resonator waveguide. The probe wavelength was adjusted to match the maximum spectral slope of a transmission resonance. The output probe wave was amplified by another EDFA to an average optical power of 1 mW and detected by a broadband photo-receiver. The detector output voltage was monitored by the input port of the network analyser or by an RF spectrum analyser.

Figure. 4a shows an example of the measured transfer function of RF voltage between the modulation of the incident pump wave and that of the detected output probe wave. The grating period $\Lambda$ in the device under test was 1.4 μm. A main transmission peak is centred at a frequency of $2.45 \pm 0.1$ GHz, and a second, weaker peak is observed at $2.85 \pm 0.1$ GHz. The two frequencies are in good agreement with those of the numerically calculated lowest-order surface-acoustic modes of the SOI layer stack, with a wavelength of 1.4 μm. The calculated spatial profiles of the two modes are shown in Fig. 4b. The full width at half maximum of the primary transmission peak is 200 MHz. The excitation bandwidth is inversely proportional to the number of periods in the grating[49], and larger gratings could give rise to narrower spectra. The useful grating size is restricted, however, by acoustic propagation losses (see also below). The magnitude of the output modulation voltage scales linearly with $P_{\text{pump}}$ (Fig. 4c).

Stimulated surface waves pass across two straight parallel waveguide sections within the race-track resonator layout: a first section that is few microns away from the gold grating, and a second section following additional acoustic propagation distance

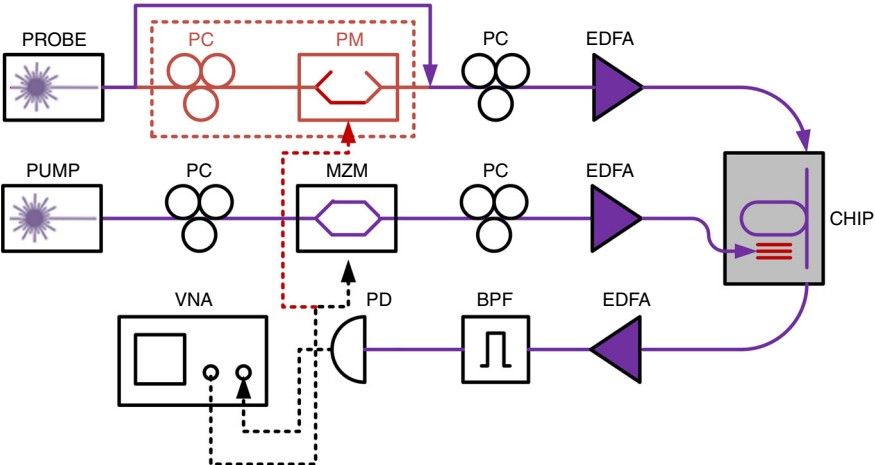

**Fig. 3** Schematic illustration of the measurement setup. Pump light from a first laser diode is amplitude-modulated in an electro-optic Mach-Zehnder modulator (MZM). The modulator is driven by a sine wave of radio-frequency from the output port of a vector network analyser (VNA). The pump wave is amplified by an erbium-doped fibre amplifier (EDFA), and the end facet of the amplifier output fibre is placed above the gold grating of a device under test. An optical probe wave from a second laser diode is amplified by a second EDFA and coupled into the input port of a race-track resonator waveguide in the same device. An electro-optic phase modulator (PM) in the probe input path is bypassed in most measurements. It serves in auxiliary experiments for estimating the magnitude of photo-elastic index perturbations (see Methods). The stimulation of surface-acoustic waves by the pump light induces amplitude modulation of the probe wave at the resonator output (see also Fig. 1). The output probe wave is amplified by another EDFA and detected by a broadband photo-detector (PD). An optical bandpass filter (BPF) suppresses the amplified spontaneous emission of the optical amplifiers. The electrical output of the photo-receiver is analysed by the input port of the VNA. The modulation radio-frequency is swept to obtain the opto-mechanical transfer function of the device under test. PC: polarization controller

$z$ of 100 μm. The two crossings are illustrated in Fig. 2a. Consequently, the probe wave within the race-track resonator undergoes two photo-elastic perturbation events in each path. Phasor addition of the two events depends on the exact radio-frequency $f$ of the acoustic wave. The transfer function of the device is therefore characterized by interference fringes (see Fig. 4d). The spectral period of the fringes pattern is 35 MHz, in agreement with the acoustic propagation delay of 28.5 ns over the distance $z$. The propagation losses of acoustic intensity are estimated as $19.5 \pm 2.5$ dB × mm$^{-1}$ based on the fringes visibility (see Methods). The losses correspond to an effective propagation length of $225 \pm 25$ μm. Figure 4d also shows the measured transfer function of the same device following the deposition of a photo-resist on top of the silicon device layer within the race-track perimeter (see image in Fig. 4e). The resist serves as an acoustic absorber, and blocks the SAWs from reaching the second crossing of the race-trace waveguide. The interference fringes visibility is reduced accordingly.

Figure 5a presents the frequencies of the lowest-order peaks in the measured transfer functions for several devices, with different spatial periods $\Lambda$ of the gold gratings. The frequencies match those of the calculated fundamental surface-acoustic modes of wavelengths $\Lambda$. The SAW frequency reaches 8 GHz for $\Lambda = 500$ nm (see Fig. 5b). The stimulation method is scalable, in principle, to even higher frequencies, using metal patterns with shorter spatial periods[46,50]. However, measurements were restricted by the bandwidths of our amplitude modulator, photo-receiver, and network analyser.

The magnitude of photo-elastic perturbations to the effective index of the optical mode in the race-track waveguide was calibrated using the following procedure: the amplitude modulation of the pump wave was switched off, and the input probe wave passed through an electro-optic phase modulator instead (see Fig. 3). The phase modulator was driven by a sine wave at the radio-frequency $f$ of interest. Propagation through the race-track resonator converts the phase modulation of the input probe to intensity modulation at the output. The drive voltage to the phase modulator was varied until the magnitude of output probe modulation equalled that induced by a SAW of the same frequency and a specific $P_{pump}$ of interest. That value of modulation voltage provides an estimate for the photo-elastic index perturbations magnitude (see Methods for details). The index modulation $\Delta n$ was $1.2 \times 10^{-6} \pm 0.3 \times 10^{-6}$ refractive index units (RIU) at $P_{pump}$ of 500 mW (or intensity of about 18 kW × cm$^{-2}$). The experimental uncertainty corresponds to the standard deviation among the measurements of three devices.

Figure 6 presents the RF power spectrum of the output voltage, obtained for $P_{pump} = 500$ mW, $f = 2.4094$ GHz, and $\Lambda = 1.4$ μm. The measurement bandwidth was 10 kHz. RF peak power of −75 dBm (voltage magnitude of 55 μV) is observed at the frequency of stimulation. The DC output voltage was 50 mV. The modulation depth of 0.11% is consistent with the above estimate for $\Delta n$ (see Methods). The ratio between the peak power and the RF noise power within 1 Hz bandwidth is 70 dB. The signal-to-noise ratio (SNR) may be enhanced with several different schemes (see Discussion below).

Lastly, the layout of the optical resonator waveguide was modified to include four parallel stretches, separated by equal spacing of 40 μm (Fig. 7a). Stimulated SAWs from a 1.4-μm-period gold grating cross the waveguide four times, with a unit differential delay of 10.6 ns between successive events. Figure 7b shows the measured normalized RF power transfer function of the device. The response matches that of a four-tap discrete-time microwave-photonic filter with equal weights. Periodic passbands are observed within the spectral envelope of the SAW stimulation response (see also Fig. 4d). The free spectral range of the filter is 95 MHz, and the full width at half maximum of the passbands is 22 MHz. A device with six crossings and a unit differential delay of 8 ns is shown in Fig. 7c. The device footprint is $200 \times 200$ μm$^2$. The longest delay difference within that device is 40 ns, corresponding to the optical delay over 8 m of fibre. The measured transfer function is in very good agreement with the

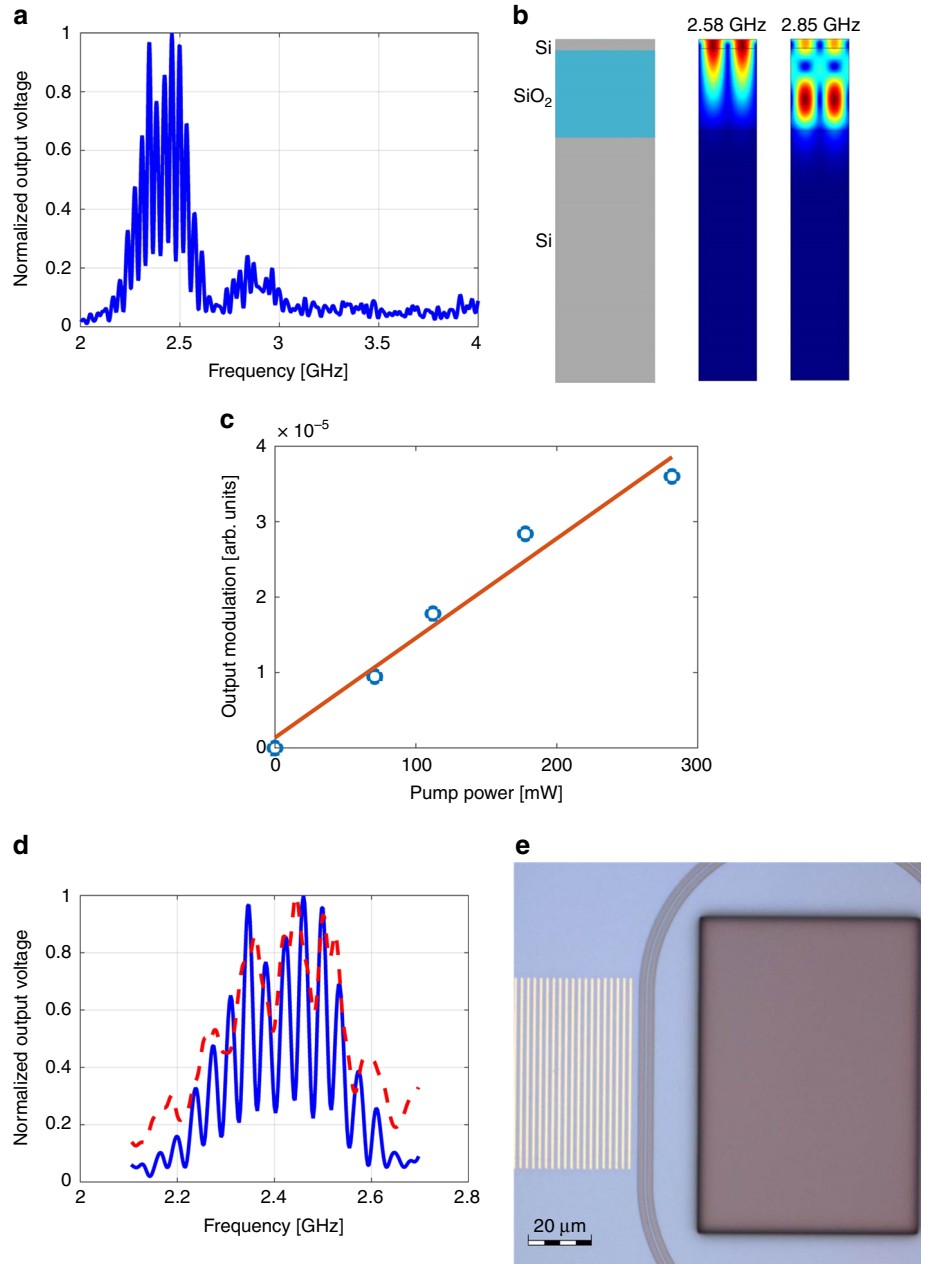

**Fig. 4** Surface-acoustic wave modulation of the output probe wave. **a** Measured normalised transfer function between the input modulation voltage of the optical pump and that of the detected output probe, as a function of radio-frequency. The pump wave illuminated a grating of thin gold stripes with a period of 1.4 μm. Two peaks correspond to stimulation of two surface-acoustic modes of the silicon on insulator layer stack. **b** Calculated spatial profiles of the two lowest-order surface-acoustic modes of the silicon on insulator layer stack, with an acoustic wavelength of 1.4 μm. The frequencies of the two modes match those of the transmission peaks of **a**. **c** Circular markers: measured output modulation voltage magnitude at 2.45 GHz frequency as a function of the average optical power of the pump wave. Solid line: a linear fit. **d** Solid blue line: magnified view of the measured normalised voltage transfer function of **a**, near the main peak. Fringes with a spectral period of 35 MHz are observed. The period corresponds to the 28.5 ns of acoustic group delay along the 100 μm separation between the two straight sections of the race-track resonator. The transfer function represents two photo-elastic modulation events of the probe wave within the race-track resonator: one in the straight waveguide section immediately adjacent to the metallic grating, and another in the second straight section 100 μm away (see also illustration in Fig. 2a). Dashed red line: same measurement following the deposition of an acoustic absorber (photo-resist) within the race-track perimeter (see a top-view microscope image in **e**. The scale bar represents 20 μm.). The surface waves are blocked from reaching the straight waveguide section at the far side of the race-track resonator. The spectral fringes visibility is reduced accordingly. Source data are provided as a Source Data file

design of a six-tap microwave-photonic filter, with a free spectral range of 125 MHz and 20 MHz-wide passbands (Fig. 7d). The results demonstrate the application of SAW-photonic devices in finite-impulse-response, integrated microwave-photonic filters in SOI.

## Discussion

This work presents a first introduction of opto-mechanical interactions and signal processing to standard SOI photonics. The devices do not involve the suspension of membranes or wave-guides, and heterogeneous integration of additional material

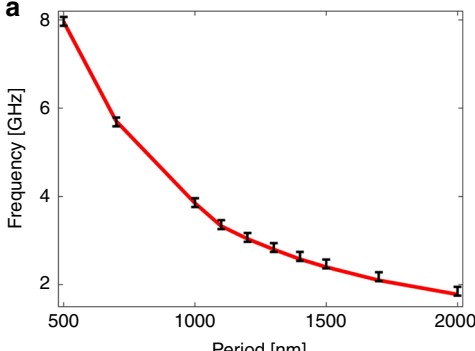
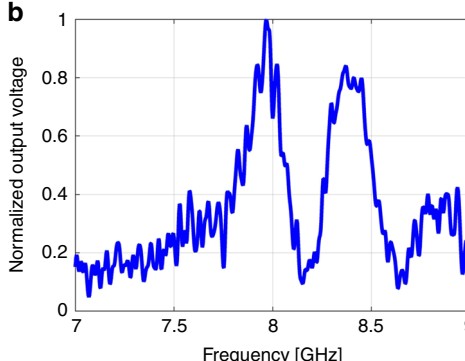

**Fig. 5** Frequencies of surface-acoustic modes. **a** Black markers: measured radio-frequencies of the lowest-order transfer function peaks vs. the spatial period of the gold gratings. The experimental uncertainty corresponds to the spectral widths of transmission peaks. Solid red trace: calculated frequencies of the fundamental surface-acoustic modes of the silicon on insulator layer stack as a function of wavelength. Good agreement between measurements and calculations is observed. **b** Measured normalised transfer function between the input modulation voltage of the optical pump and that of the detected output probe, as a function of radio-frequency. The pump wave illuminated a grating of thin gold stripes with a period of 500 nm. Transmission peaks at 8 and 8.4 GHz correspond to the lowest-order surface-acoustic modes of the silicon on insulator layer stack at 500 nm wavelength. Source data are provided as a Source Data file

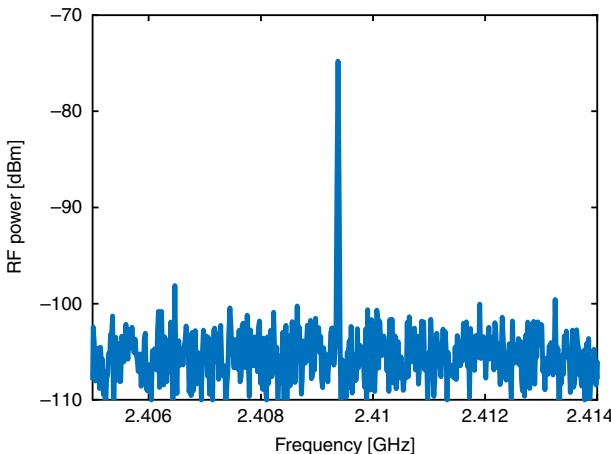

**Fig. 6** Measured radio-frequency spectrum of output probe voltage. Measurements were taken with a bandwidth of 10 kHz. A surface-acoustic wave at 2.4094 GHz frequency was stimulated by a 500 mW pump wave and a 1.4-μm-period gold grating. The peak power is −75 dBm. The ratio between the power of the peak and that of the background noise within 1 Hz bandwidth is 70 dB. Source data are provided as a Source Data file

platforms is not required. Light waves propagate in standard ridge waveguides. As an alternative to piezo-electric actuation, SAWs are stimulated through the absorption of pump light in periodic metallic patterns and subsequent thermo-elastic expansion. This method of SAWs stimulation is generic and applicable, in principle, to any substrate. The frequency of the stimulated acoustic waves may be chosen arbitrarily: the excitation of 90 GHz waveforms has been reported in bulk substrates[46,50].

The surface waves stimulation principle was previously implemented through the absorption of ultra-short pump pulses in two-dimensional metal structures[44–46]. However, most of the spectral contents of the short pulses do not couple to the acoustic mode. In contrast, we use continuous pump light that is modulated at the radio-frequency of interest. This pumping scheme provides more efficient coupling to the surface wave. The average pump power used in this work is an order of magnitude lower than those of previously used ultra-short pulses[44–46]. In addition, the monitoring of SAWs in previous works had relied on complex

free-space measurements of the reflection of an off-plane probe wave[44–46]. Here, photo-elastic modulation of probe waves in planar photonic integrated circuits is used instead.

The concept of SAW-based RF signal processing was successfully carried over from the realm of analogue electronics, where it is known for over 50 years, into integrated photonics. SAWs were successfully used in modulation and switching of photonic circuits in GaAs[8–10], however the objectives of these works did not include time delay and filtering of RF signals. The delay of microwave signals by 40 ns was realized on-chip within 150 μm (Fig. 7). The delay elements pave the way towards analogue opto-mechanical signal processing in standard SOI. As a first application example, discrete-time integrated microwave-photonic filters with four and six taps, free spectral ranges up to 125 MHz and 20 MHz-wide passbands were demonstrated. The complex weights of individual taps can be chosen arbitrarily, through the exact position, width and curvature of resonator waveguide sections. To the best of our knowledge, the devices represent a first realization of discrete-time microwave-photonic filters based on acoustics.

The photo-elastic index perturbations obtained in this work are comparatively weak: $1.2 \times 10^{-6}$ RIU for a pump wave intensity of $18 \, \text{kW} \times \text{cm}^{-2}$. The intensity modulation depth of the output probe wave is restricted to 0.1%, and the output SNR over the 20 MHz bandwidth of integrated microwave-photonic filters is presently on the order of unity: a value too low for many signal processing applications. However, the current device metrics do not represent restrictive upper limits on the performance of proposed SAW-photonic concept. Multiple solution paths are available to enhance the output signals by orders of magnitude.

First, the photo-thermal stimulation of SAWs can be made more efficient. The composition and thickness of the metal stripes may be optimized to induce larger strain in the SOI surface. The embedding of the grating stripes within shallow etched patterns in the silicon layer may enhance the SAWs generation[51]. Concentric metal rings may be used instead of parallel grating stripes to focus the surface waves to a few-microns-size spot, in which the SAW magnitude could be 10-fold stronger[7–10,52]. Furthermore, acoustic reflectors can be patterned in the silicon device layer to the sides of the probe resonator waveguide[32], to provide feedback cavity enhancement of the surface waves[32].

Even for a given SAW magnitude, the intensity modulation depth of the probe wave may be increased significantly. The modulation depth is proportional to the quality factor of the

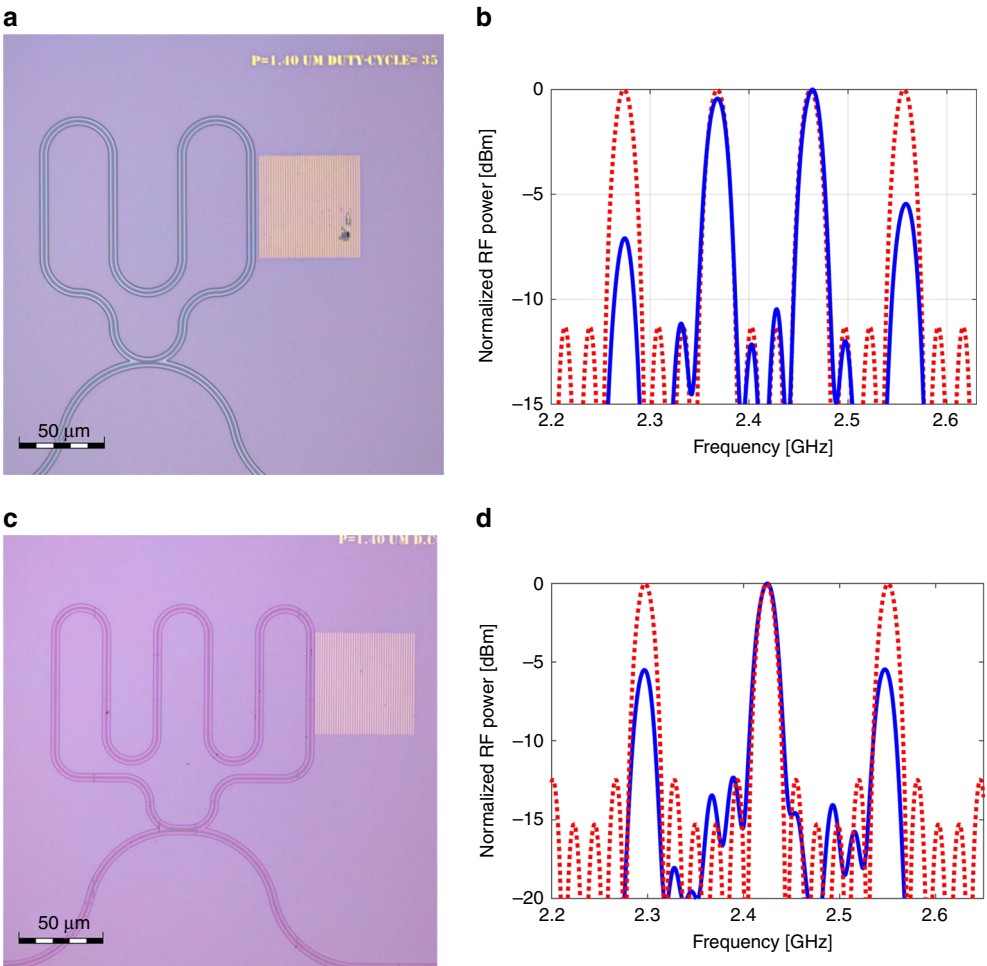

**Fig. 7** Multi-tap discrete-time integrated microwave-photonic filters. **a** Top-view image of a device with four straight sections within the optical resonator waveguide. The stretches are separated by equal spacing of 40 μm, corresponding to a unit acoustic differential delay of 10.6 ns. A gold grating with period of 1.4 μm is defined near the resonator. The scale bar represents 50 μm. **b** Measured normalised magnitude transfer function between the modulation voltage of the optical pump and that of the detected output optical probe, as a function of radio-frequency (solid, blue). Periodic passbands are observed within the spectral envelope of the SAW stimulation (see also Fig. 4d). The free spectral range of the filter response is 95 MHz, and the full width at half maximum of the filter passbands is 22 MHz. The dashed red trace shows the calculated transfer function of a four-tap, discrete-time microwave-photonic filter with equal complex weights for all taps and a unit group delay of 10.6 ns. Close agreement between the filter design and the measured response is observed. **c** Top-view image of a device with six straight waveguide sections, separated by 30 μm (8 ns). The scale bar represents 50 μm. **d** Measured normalised magnitude transfer function of the device shown in **c** (solid blue line), alongside the calculated transfer function of a six-tap microwave-photonic filter with equal complex weights and a unit differential group delay of 8 ns (dashed red line). The free spectral range and passbands full width at half maximum are 125 and 20 MHz, respectively. Good agreement is obtained again between design and experiment. Source data are provided as a Source Data file

probe wave resonator (see Methods). Devices available to us had a modest loaded quality factor of only 40,000, whereas the state of the art for SOI is ten times better. The modulation depth also scales with the relative fraction of the resonator cavity length that is affected by acoustic waves (see Methods). In present devices, only 60-μm-long sections within a 480-μm-long race-track resonator are perturbed by SAWs. That lengths ratio can be brought close to 100% if the resonator layout is replaced, for example, by a local defect cavity along a Bragg grating in an SOI waveguide[52].

Lastly, the output voltage and SNR may also be enhanced even if the modulation depth of the probe wave remains the same. The SNR is currently restricted by the amplified spontaneous emission of the fibre amplifier at the probe wave output path. Optical gain compensates for comparatively large coupling losses of the probe wave in and out of devices: about 10 dB per facet. Coupling losses as low as 2–3 dB per interface are reported in the literature[53]. Better coupling would provide the same output power with

weaker gain, leading to lower amplifier noise. The magnitude of the output voltage signal is presently limited by the maximum power handling of a standard photo-diode. Stronger probe waves may be accommodated using high-power detectors, which are widely employed in microwave-photonics applications[32,54].

While the strength of photo-thermal excitation of SAWs would not match that of piezo-electric actuation, future extensions of the proposed concept may well meet the requirements of microwave-photonic signal processing applications. The 200-μm propagation length of 2.5 GHz SAWs in SOI is sufficient for tens of nanoseconds of acoustic group delay, and can accommodate tens of optical waveguide taps in a spiral resonator layout. In addition to potential use in signal processing, the technique is also suitable for studying the acousto-optic properties of materials, and for sensing applications[50,52].

In summary, SAW-photonic devices may add another dimension to integrated microwave-photonics, especially in

silicon. Building blocks may be brought together with hybrid InP on SOI light sources, silicon-photonic free-carrier modulators and germanium on silicon detectors to form entire systems[47,48]. Such integration of analogue photonic processing is widely regarded as an enabling technology for next-generation (5G) cellular networks[55]. Ongoing work is being dedicated to the application of the concept in larger signal processing modules.

## Methods

**Fabrication of devices**. Devices were fabricated in standard SOI wafers with a 220-nm-thick silicon device layer on top of a 2-μm-thick buried oxide layer. Gratings of periodic gold stripes were defined by electron-beam lithography, sputtering of a 5-nm-thick chromium adhesion layer followed by 20 nm of gold, and a lift-off process. The sputtering rates for chromium and gold were 0.2 nm × s$^{-1}$ and 0.5 nm × s$^{-1}$, respectively. The vacuum level during sputtering was $5 \times 10^{-6}$ bar and the cycle rate was 5 rpm. Optical waveguides were defined in the silicon device layer using a second phase of electron-beam lithography, followed by inductively coupled plasma reactive-ion etching. Alignment markers were used to define the locations of optical waveguides with respect to the gold gratings. The etching process used a mixture of SF$_6$ and C$_4$F$_8$ gasses, at flow rates of 65 and 10 ccm, respectively. Etching was carried out at a vacuum level of $4 \times 10^{-10}$ bar, RF power of 100 W and 6 nm × s$^{-1}$ rate. Ridge waveguides were partially etched to a depth of 70 nm. The width of the ridges was 700 nm. Vertical grating couplers were patterned at the ends of the bus waveguides of race-track resonators. In several devices, 2.5-μm-thick patches of AZ1518 photo-resist were patterned inside the race-track perimeter through photo-lithography. The resist patches served as acoustic absorbers.

**Estimate of surface-acoustic waves propagation losses**. Consider SAWs that are stimulated in a metallic grating adjacent to a ring resonator, as shown in Fig. 2a. The surface waves first path through one straight section of the race-track layout, introducing photo-elastic index perturbation of magnitude $\Delta n$. Following a propagation distance $z = 100$ μm, the SAWs cross a second straight section of the resonator (Fig. 2a). The magnitude of photo-elastic perturbation in that second waveguide segment is $\Delta n \exp(-\alpha_{SAW} z/2)$, where $\alpha_{SAW}$ denotes the propagation losses coefficient of SAWs intensity in units of m$^{-1}$. Perturbations in both sections contribute to intensity modulation of the output probe wave. Their addition is constructive (destructive) when the propagation distance $z$ is an even (odd) integer multiple of half the surface-acoustic wavelength. The transfer function of the device is therefore characterized by a fringes pattern (see Fig. 4d). The fringes visibility is given by $[1 + \exp(-\alpha_{SAW} z/2)]/[1 - \exp(-\alpha_{SAW} z/2)]$. The measured visibility of 4 ± 0.5 corresponds to $\alpha_{SAW} = 19.5 \pm 2.5$ dB × mm$^{-1}$.

**Calibration of photo-elastic index perturbation magnitude**. Let us denote the instantaneous photo-elastic perturbation to the effective modal index of the race-track resonator waveguide as $\Delta n(f)\sin(2\pi ft)$, where $f$ is the radio-frequency of a SAW and $t$ stands for time. The index perturbations induce periodic variations in the single-path phase delay of a probe wave within the resonator: $\Delta \varphi = k_0 l \Delta n(f)\sin(2\pi ft)$. Here $k_0$ is the vacuum wavenumber of the probe wave and $l$ denotes the length of the waveguide section that is affected by the SAW, which approximately equals the length of the stripes in the gold grating.

The variations in the optical phase delay introduce oscillating spectral shifts in the transfer function of the resonator, with respect to the fixed optical frequency of the probe wave. For example, phase delay variations of $2\pi$ correspond to a spectral offset by a single free spectral range: $c/(nL)$, where $c$ is the speed of light in vacuum, $n$ denotes the group index of the race-track waveguide, and $L$ is the race-track length. SAWs introduce phase variations $\Delta \varphi$ that are much smaller than $2\pi$. The magnitude of spectral shifts in the transfer function is given by:

$$\Delta \nu_{SAW}(f) = \frac{k_0 l \cdot \Delta n(f)}{2\pi} \frac{c}{nL} = \nu_0 \frac{l}{L} \frac{\Delta n(f)}{n}. \tag{1}$$

Here $\nu_0$ stands for the fixed optical frequency of the probe wave (~193 THz), which is chosen on a spectral slope of the resonator transfer function. The spectral frequency shifts of the resonator response manifest in amplitude modulation of the output probe.

In order to calibrate the magnitude of index perturbations, auxiliary measurements were carried out with the intensity modulation of the pump wave switched off. Instead, the probe wave at the input end of the race-track resonator passed through an electro-optic phase modulator (see Fig. 3). The phase modulator was driven by a sine wave voltage at radio-frequency $f$ and magnitude $V_{PM}(f)$. The phase modulation corresponds to periodic shifts in the instantaneous optical frequency $\nu(t)$:

$$\nu(t) = \nu_0 + \pi \frac{V_{PM}(f)}{V_\pi} f \cdot \cos(2\pi ft) = \nu_0 + \Delta \nu_{PM}(f) \cdot \cos(2\pi ft). \tag{2}$$

Here we denote the magnitude of the frequency modulation of the probe wave as $\Delta \nu_{PM}(f) \equiv \pi f V_{PM}(f)/V_\pi$, and $V_\pi \sim 3.5$ V is the voltage required for an electro-optic phase shift of $\pi$ radians. Similar to the photo-elastic perturbations discussed

above, the frequency modulation of the input probe wave is converted to intensity modulation at the output end. Here, however, the instantaneous optical frequency of the probe is shifting with respect to a stationary resonator response, rather than the other way around.

In the experiment, $V_{PM}(f)$ is modified until the voltage modulation of the detected output probe matches the level previously measured for SAWs at the pump power of interest. Both measurements are taken for the same $\nu_0$ and the same probe power. Matching between the magnitudes of the output voltage modulation in the two cases signifies $\Delta \nu_{PM}(f) = \Delta \nu_{SAW}(f)$, leading to the following estimate for the photo-elastic index perturbations magnitude:

$$\frac{\Delta n(f)}{n} = \pi \frac{V_{PM}(f)}{V_\pi} \frac{f}{\nu_0} \frac{L}{l}. \tag{3}$$

**Modulation depth of the output probe**. The conversion of photo-elastic index perturbation to probe intensity modulation is determined by the spectral slope of the resonator power transfer function $T(\nu_0)$. The exact value of the derivative $\partial T(\nu_0)/\partial \nu_0$ depends on the precise tuning of $\nu_0$. However, for a resonator close to critical coupling, we may approximate $\partial T(\nu_0)/\partial \nu_0 \approx 1/\Delta \nu_{FWHM}$ where $\Delta \nu_{FWHM}$ is the full width at half maximum of $T(\nu_0)$. Using Eq. (1), we obtain an estimate for the magnitude $\Delta P_{Pr}(f)$ of the modulation of the probe output power:

$$\frac{\Delta P_{Pr}(f)}{P_{Pr}} \approx \frac{\Delta \nu_{SAW}(f)}{\Delta \nu_{FWHM}} = \frac{\Delta n(f)}{n} \frac{l}{L} \frac{\nu_0}{\Delta \nu_{FWHM}} = \frac{\Delta n(f)}{n} \frac{l}{L} Q. \tag{4}$$

Here $Q$ is the loaded quality factor of the race-track resonator and $P_{pr}$ denotes the average output power of the probe wave. For the parameters of this work: $Q \sim 40,000$, $L = 480$ μm, $l \sim 60$ μm, $n = 3.5$ RIU and $\Delta n \sim 1.2 \times 10^{-6}$ RIU, the modulation depth $\Delta P_{Pr}(f)/P_{Pr}$ of the output probe is expected to be on the order of 0.17%. This estimate is in agreement with measurements.

## Data availability
The source data underlying Figs. 2b, 4a, c, d, 5a, b, 6, 7b, d are provided as a Source Data file.

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

## Acknowledgements

This work was supported in part by a Starter Grant from the European Research Council (ERC), grant number H2020-ERC-2015-STG 679228 (L-SID).

## Author contributions

D.M. designed the devices and performed simulations. M.K. and M.H. led the fabrication efforts. D.M. and M.K. performed the optical and acousto-optic characterization of devices. M.P., M.F., and T.S. participated in the fabrication of devices. M.H. and M.P. carried out the microscopy analysis of devices. S.L. proposed the photo-elastic modulation of resonator waveguides. A.B. assisted with the assembly of the measurement setup and the literature survey. A.Z. proposed the concept of surface-acoustic waves-photonics in silicon on insulator, managed the project, and wrote the paper.

## Additional information

**Competing interests:** The authors declare no competing interests.

**Peer Review Information** *Nature Communications* thanks Jose Azana and other, anonymous, reviewer(s) for their connotestribution to the peer review of this work. Peer reviewer reports are available.

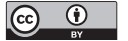

