## [Peer Review File · Nature Communications]

Reviewers' comments:

Reviewer #1 (Remarks to the Author):

Munk et al. demonstrate photothermal stimulation of surface acoustic wave (SAW) on the SOI platform to modulate integrated photonic devices. They further use the principle to realize multi-tap microwave photonic filters. While there has been a large amount of prior work on using SAW to modulate photonic devices, the innovation of the current work is to achieve that on the standard SOI platform without using piezoelectric materials. However, the drawback of this technique and the SOI platform is obvious: 1) The efficiency of photothermal excitation of SAW is very low; 2) an additional EO modulator is required to convert RF signal to the optical signal; 3) the propagation loss of SAW on SOI is very high. Therefore, the demonstrated photothermal excitation of SAW using an integrated metal grating is only an interesting technique but without the promise to a realistic technology. Thus, this reviewer does not support publication in Nature Communications.

Comments:

1. The following important and close related prior works are not cited:

M. Beck, M. M. de Lima, E. Wiebicke, W. Seidel, R. Hey, and P. V. Santos, "Acousto-optical multiple interference switches," (in English), *Applied Physics Letters*, vol. 91, no. 6, 061118, Aug 6 2007.

M. Beck, M. M. de Lima, and P. V. Santos, "Acousto-optical multiple interference devices," *Journal of Applied Physics*, vol. 103, no. 1, 014505, Jan 1 2008.

M. de Lima Jr, M. Beck, R. Hey, and P. Santos, "Compact Mach-Zehnder acousto-optic modulator," *Applied physics letters*, vol. 89, no. 12, 121104, 2006.

2. There are too many figures. They should be combined into multiple panels.

3. In Fig. 6, SAW frequency up to 8 GHz is excited but not spectral data is included.

4. The purpose of Fig. 7 is unclear. It doesn't contain much useful information.

5. For RF/microwave filtering, the insertion loss is very important. A few dB is considered too high. Clearly, for this purpose, the current device is very far from realistic application.

Reviewer #2 (Remarks to the Author):

This paper reports what I believe to be the first SAW device implementation on an integrated SOI platform. This is a very significant and exciting advancement because this may enable integration of the wide range of important SAW-based functionalities directly in SOI chips, arguably the most prominent material platform for future integrated photonics devices and systems. The proposed technology is surprisingly simple and practical as it avoids piezo-electric actuators, suspension of waveguides or hybrid material integration.

The central idea is to use a gold grating with the suitable period on top of the SOI substrate. This enables excitation of a SAW by pumping the grating with an optical lightwave amplitude modulated by an RF sine wave. This mechanism was previously employed on bulk silicon wafers but it has been successfully transferred to an SOI device here for the first time. This is not an obvious step. To have a working device, detection is achieved through a probe optical field that will be modulated by the SAW as this interacts with a properly designed SOI optical resonator. I think this is a very original and neat strategy and it is convincingly demonstrated in the paper through solid experimental work. Moreover, the authors have succeeded in using this idea for demonstration of microwave-photonic filters with a performance that is enhanced by the SAW interaction in ways that may be very challenging to achieve otherwise. For instance, the mechanism has been used for demonstration of delay lines over tens of nanoseconds and GHz bandwidths by taking advantage

of the inherent slow speed of the propagating SAW, i.e., using a hundreds micron length only. This is very impressive and it provides a convincing proof of the interesting research opportunities that are opened by the reported work.

Overall, I think the reported work has the standards that one may expect for a publication in a Nature journal. The paper is well written and a pleasure to read, with a good organisation of the presented material, from the introduction of the central ideas to the demonstrations of the basic principles and applications. Still, the following aspects should be clarified before the paper can be accepted for publication:

1- The frequency response in Fig. 4 shows that the SAW is excited over a relatively large bandwidth (hundreds of MHz). Depending on the target application, one may want to excite a narrower bandwidth of SAW frequencies. Discussions should be provided on the origin of this 'operation' bandwidth and potential mechanisms available to the designer for customising (e.g., narrowing) this bandwidth, if possible.

2- I have had difficulties to understand in a first pass the origin of the oscillations in the measured spectral response, Figs. 4 and 5. I would suggest the authors to try to improve the corresponding explanations in the paper, e.g., concerning the text on Fig. 5(a). It may help adding some well-thought plot to illustrate how the probe wave is modulated 'twice' with a well-defined time delay in between the modulations, leading to the formation of the observed spectral fringes. This is important because it should also help in picturing the relationship of this process with the delayed taps that are later on utilised for design of microwave-photonic filters.

Reviewer #3 (Remarks to the Author):

In their paper "Surface Acoustic Wave – Photonic Devices in Silicon-on-Insulator" the authors present SAW-photonic devices in standard SOI, avoiding the need for piezoelectric actuation. Incident RF waveforms at Gigahertz frequencies are converted from the modulation of pump light to the form of surface waves, and then to the modulation of an optical probe that is propagating in a standard photonic circuit. Acoustic delays of tens of nanoseconds are achieved on-chip. The results demonstrate the application of SAW-photonic devices in finite-impulse-response, integrated microwave-photonic filters in SOI.

The results here presented are novel and of interest both for the photoacoustic and opto-electronics community.

The results are convincing and extended evidences are provided supporting the claims. The paper is well organized. The authors did a good job in avoiding unnecessary details while preserving enough information so as to allow the reader to reproduce the results. This good balance makes the reading enjoyable.

In my opinion the paper will be of impact both in the opto-electronics and photoacoustic community. As for the former, the application of hypersonic SAW-photonic devices in finite-impulse-response, integrated microwave-photonic filters in SOI will boost high frequency SAW applications. As for the latter, I agree with the authors claim:

"However, most of the spectral contents of the short pulses do not couple to the acoustic mode. In contrast, we use continuous pump light that is modulated at the radio-frequency of interest. This pumping scheme provides more efficient coupling to the surface wave. The average pump power used in this work is an order of magnitude lower than those of previously-used ultra-short pulses 41-43. In addition, the monitoring of SAWs in previous works had relied on complex free-space measurements of the reflection of an off-plane probe wave 41-43. Here, photo-elastic modulation of probe waves in planar photonic integrated circuits is used instead."

and find their proposal a good alternative to impulsive photoacoustic generation and time-resolved read out via pump&probe techniques.

I also liked the way the author calibrated the magnitude of photo-elastic perturbations to the effective index of the waveguide. This method may prove useful in characterizing the acousto-optic conversion of optical materials, a topic of impact both for the photoacoustic and opto-electronic community

The bibliography should be expanded in order to support some claims. Specifically:

1. On page 5, where the authors claim that:

"These metrics can be improved significantly in future work. The index modulation may be increased by an order of magnitude through focusing of the acoustic waves into micron-scale regions 7 , such as defect cavities within Bragg gratings.", they should cite the paper from Giannetti et al. IEEE Photonics Journal 1, 21 (2009).

This paper proposes a mean to localize GhZ acoustic waves in a Si substrates.

2. On page 4, where the authors claim:

"The stimulation method is scalable, in principle, to even higher frequencies, using metal patterns with shorter spatial periods 43", reference should be made to the possibility of using metallic array period down to tens of nm in order to trigger acoustic waves in the 100 Ghz range, see the review from:

Nardi et al., IEEE SENSORS JOURNAL, VOL. 15, NO. 9 (2015).

3. The same reference as in point 2 should be mentioned when the authors state, on page 5, that:

"The excitation of 90 GHz waveforms has been reported in bulk substrates 43

For the above mentioned reasons I believe the manuscript fulfills the requirements required for publications on Nature Communications and recommend publication of the manuscript once:

1. the bibliography will be expanded
2. the minor point hereafter reported will be taken care of

Minor points:

Fig. 1 periods match those OF a surface acoustic

Figure 2 (a) the scale is hardly visible.

Figure (5): "The propagation losses of acoustic intensity for the fundamental surface mode are estimated as 40 ± 5 dB/mm based on the fringes visibility."

Please expand in methods how the "estimate" is performed.

Fig. 6.: Orange marks hardly visible.

"of thee silicon-on insulator layer stack."

Authors Reply to Reviews

The following presents our reply to the reviews of our manuscript: "Surface Acoustic Wave – Photonic Devices in Silicon-on-Insulator"

Response to Reviewer 1

"Munk et al. demonstrate photothermal stimulation of surface acoustic wave (SAW) on the SOI platform to modulate integrated photonic devices. They further use the principle to realize multi-tap microwave photonic filters. While there has been a large amount of prior work on using SAW to modulate photonic devices, the innovation of the current work is to achieve that on the standard SOI platform without using piezoelectric materials. However, the drawback of this technique and the SOI platform is obvious: 1) The efficiency of photothermal excitation of SAW is very low; 2) an additional EO modulator is required to convert RF signal to the optical signal; 3) the propagation loss of SAW on SOI is very high. Therefore, the demonstrated photothermal excitation of SAW using an integrated metal grating is only an interesting technique but without the promise to a realistic technology. Thus, this reviewer does not support publication in Nature Communications."

Reply: We are happy that the reviewer finds innovation and interest in the introduction of SAWs to the standard SOI platform, without use of piezo-electric materials. We agree with the reviewer that much progress is still required for real-world applications of the proposed concept. While such progress cannot be guaranteed, we do not share the reviewer's conviction that the principle is "without promise of realistic technology". **On top of conceptual scientific novelty, we already demonstrate in this work a first example of integrated microwave photonic filters using SAWs on SOI.** Such filters are difficult to implement otherwise (see also a like-minded comment by Reviewer 2 below). In addition, the current device metrics should not be regarded as restrictive upper limits on the performance of the proposed SAW-photonic platform. Multiple avenues are open for potential performance enhancement:

- a. **Stronger SAWs excitation.** We accept that photo-thermal stimulation of SAWs is much weaker than piezoelectric actuation. We made the same comment ourselves in the Discussion Section. That being said, the magnitude of stimulated SAWs can be increased beyond the present results. The thickness and composition of the metallic grating stripes can be optimized. Shallow etching of the silicon layer prior to metal deposition can enhance SAWs stimulation [51]. Furthermore, concentric metal rings may be used instead of parallel grating stripes to focus the SAWs to a few-microns-size spot, in which perturbations would be ten-fold stronger (see for example the new references [8-10] that were pointed out by the reviewer).
- b. **Deeper modulation of the probe wave intensity for given excitation.** As noted in Eq. (4), the modulation depth of the probe wave scales with the Q factor of the resonator. Devices available to us had a modest Q factor of only 40,000. The state of the art in silicon photonics is ten times better. The incorporation of higher-Q resonators is already under study. In addition, modulation depth also scales with the spatial overlap between the SAW wave-front and the resonator length (see Eq. (4) again). This overlap was only 10-20% in the race-track layout used. However, the overlap can be brought close to 100% if the probe resonator is replaced with a point defect in a Bragg grating along a silicon waveguide [50]. This solution path is under study as well. Lastly, etched features to the sides of the probe waveguide can provide acoustic cavity enhancement of the SAW magnitude at the waveguide region, as reported in suspended silicon membranes [32]. **The probe modulation depth can be at least an order of magnitude larger than the present value, even if the strength of the SAWs is unchanged.**
- c. **Larger output voltage modulation and/or higher SNR for given modulation depth.** The SNR of the output voltage (now Fig. 6) is currently restricted by the amplified spontaneous emission of a fiber amplifier at the probe wave output path. Gain is necessary to compensate for comparatively large coupling losses of the probe wave in and out of the device under test: about 10 dB per facet. Coupling losses as low as 2-3 dB per interface are reported in the literature [53]. Better coupling would provide the same output power with weaker gain, leading to lower amplifier noise. The magnitude of the output voltage signal is currently limited by the maximum power handling of a standard photo-diode. Stronger probe waves may be accommodated using

high-power photo-diodes, which are being developed for microwave-photonics applications [54]. For example, the output probe wave in a Brillouin-based microwave-photonics filter in silicon is optically amplified to 75 mW at the input of a high-power detector [32]. Therefore, **larger output signals and/or better SNR can be reached even before the SAW excitation and the modulation depth are improved.**

Regarding the need for electro-optic modulation at the input end of the pump wave, one should note that microwave-photonics links often involve the distribution of signals over long reaches of fiber. Incident microwave signals of interest are often carried on top of an optical wave to begin with. The use of electro-optic modulation is inherent to microwave photonics, and does not represent a substantial drawback in that field. We agree with the reviewer that the propagation lengths of high-frequency SAWs are limited. We even provided an experimental estimate for SAW losses at 2.5 GHz. However, the observed propagation lengths of about 200 microns are sufficient to provide tens of nanoseconds of acoustic group delay, and accommodate tens of optical waveguide taps in a spiral resonator layout. Therefore, the available propagation length may well support microwave-photonics applications. Lastly, **SAW-photonics may find additional applications beyond signal processing, such as acousto-optic characterization** (as also brought up by Reviewer 3) and sensors [50].

To summarize our reply to this comment, **we identify multiple realistic solution paths for stronger SAW excitation, deeper probe wave modulation, larger output signals and better SNR using the proposed concept.** The Discussion Section of the revised manuscript has been extended to include the above considerations.

Comment 1: "The following important and close related prior works are not cited:

M. Beck, M. M. de Lima, E. Wiebicke, W. Seidel, R. Hey, and P. V. Santos, "Acousto-optical multiple interference switches," (in English), Applied Physics Letters, vol. 91, no. 6, 061118, Aug 6 2007.

M. Beck, M. M. de Lima, and P. V. Santos, "Acousto-optical multiple interference devices," Journal of Applied Physics, vol. 103, no. 1, 014505, Jan 1 2008.

M. de Lima Jr, M. Beck, R. Hey, and P. Santos, "Compact Mach-Zehnder acousto-optic modulator," Applied physics letters, vol. 89, no. 12, 121104, 2006."

Reply: We thank the reviewer for bringing these significant and highly relevant works to our attention. We address these papers in the Introduction and Discussion Sections of the revised manuscript.

Comment 2: "There are too many figures. They should be combined into multiple panels."

Reply: We grouped together figures 4 and 5 in the revised manuscript.

Comment 3: "In Fig. 6, SAW frequency up to 8 GHz is excited but not spectral data is included."

Reply: We added a panel of the measured transfer function of a device with a 500 nm grating period (now Fig. 5(b)). Resonant excitation of surface acoustic waves at 8 GHz frequency is evident, in agreement with expectations.

Comment 4: "The purpose of Fig. 7 is unclear. It doesn't contain much useful information."

Reply: Figure 7 (now Fig. 6) is included to show the SNR of a microwave tone at the output of the device. The SNR is a key metric of device performance. In our opinion, the measured output radio-frequency spectrum shown in the figure helps the reader visualize the output SNR.

Comment 5: "For RF/microwave filtering, the insertion loss is very important. A few dB is considered too high. Clearly, for this purpose, the current device is very far from realistic application."

Reply: We agree with the reviewer that RF losses through the device are currently very high. However, these losses are not fundamental. As noted above, there is good potential to reduce RF losses by orders of magnitude through stronger SAWs stimulation, resonant enhancement and/or focusing of SAWs, deeper modulation of the probe wave, more efficient optical coupling of the probe wave, optical amplification at the output and use of high-power photo-diodes. The latter solution, in particular, is

widely employed in microwave photonics [32,54]. Prospects for reaching stronger output signals are addressed in the Discussion Section.

Response to Reviewer 2

"This paper reports what I believe to be the first SAW device implementation on an integrated SOI platform. This is a very significant and exciting advancement because this may enable integration of the wide range of important SAW-based functionalities directly in SOI chips, arguably the most prominent material platform for future integrated photonics devices and systems. The proposed technology is surprisingly simple and practical as it avoids piezo-electric actuators, suspension of waveguides or hybrid material integration."

"The central idea is to use a gold grating with the suitable period on top of the SOI substrate. This enables excitation of a SAW by pumping the grating with an optical lightwave amplitude modulated by an RF sine wave. This mechanism was previously employed on bulk silicon wafers but it has been successfully transferred to an SOI device here for the first time. This is not an obvious step. To have a working device, detection is achieved through a probe optical field that will be modulated by the SAW as this interacts with a properly designed SOI optical resonator. I think this is a very original and neat strategy and it is convincingly demonstrated in the paper through solid experimental work. Moreover, the authors have succeeded in using this idea for demonstration of microwave-photonic filters with a performance that is enhanced by the SAW interaction in ways that may be very challenging to achieve otherwise. For instance, the mechanism has been used for demonstration of delay lines over tens of nanoseconds and GHz bandwidths by taking advantage of the inherent slow speed of the propagating SAW, i.e., using a hundreds micron length only. This is very impressive and it provides a convincing proof of the interesting research opportunities that are opened by the reported work."

Reply: We thank the reviewer for his/her positive judgment of our work and for his/her support.

"Overall, I think the reported work has the standards that one may expect for a publication in a Nature journal. The paper is well written and a pleasure to read, with a good organisation of the presented material, from the introduction of the central ideas to the demonstrations of the basic principles and applications. Still, the following aspects should be clarified before the paper can be accepted for publication:"

Comment 1: *"The frequency response in Fig. 4 shows that the SAW is excited over a relatively large bandwidth (hundreds of MHz). Depending on the target application, one may want to excite a narrower bandwidth of SAW frequencies. Discussions should be provided on the origin of this 'operation' bandwidth and potential mechanisms available to the designer for customising (e.g., narrowing) this bandwidth, if possible."*

Reply: The excitation bandwidth of the surface acoustic waves is determined by the number of periods in the metallic gratings. Larger gratings with more periods could lead to narrower excitation spectra. However, the size of the grating is restricted by the propagation losses of the acoustic waves. For 2.5 GHz SAWs in SOI, that length is on the order of 200 microns. The bandwidth may therefore be reduced by about a factor of three. Lower-frequency surfaces wave can be excited through even larger grating patterns, resulting in further bandwidth reduction. This explanation was added to the Results Section.

Comment 2: *"I have had difficulties to understand in a first pass the origin of the oscillations in the measured spectral response, Figs. 4 and 5. I would suggest the authors to try to improve the corresponding explanations in the paper, e.g., concerning the text on Fig. 5(a). It may help adding some well-thought plot to illustrate how the probe wave is modulated 'twice' with a well-defined time delay in between the modulations, leading to the formation of the observed spectral fringes. This is important because it should also help in picturing the relationship of this process with the delayed taps that are later on utilised for design of microwave-photonic filters."*

Reply: We accept the comment and thank the reviewer for pointing this out. An Illustration of SAWs crossing the race-track resonator in multiple waveguide stretches was added in Fig. 2(a). The descriptions in the Results Section and the Fig. 4 Caption were rephrased for better clarity.

Response to Reviewer 3

"In their paper "Surface Acoustic Wave – Photonic Devices in Silicon-on-Insulator" the authors present SAW-photonic devices in standard SOI, avoiding the need for piezoelectric actuation. Incident RF waveforms at Gigahertz frequencies are converted from the modulation of pump light to the form of surface waves, and then to the modulation of an optical probe that is propagating in a standard photonic circuit. Acoustic delays of tens of nanoseconds are achieved on-chip."

"The results demonstrate the application of SAW-photonic devices in finite-impulse-response, integrated microwave-photonic filters in SOI. The results here presented are novel and of interest both for the photoacoustic and opto-electronics community. The results are convincing and extended evidences are provided supporting the claims. The paper is well organized. The authors did a good job in avoiding unnecessary details while preserving enough information so as to allow the reader to reproduce the results. This good balance makes the reading enjoyable."

"In my opinion the paper will be of impact both in the opto-electronics and photoacoustic community. As for the former, the application of hypersonic SAW-photonic devices in finite-impulse-response, integrated microwave-photonic filters in SOI will boost high frequency SAW applications. As for the latter, I agree with the authors claim: 'However, most of the spectral contents of the short pulses do not couple to the acoustic mode. In contrast, we use continuous pump light that is modulated at the radio-frequency of interest. This pumping scheme provides more efficient coupling to the surface wave. The average pump power used in this work is an order of magnitude lower than those of previously-used ultra-short pulses 41-43. In addition, the monitoring of SAWs in previous works had relied on complex free-space measurements of the reflection of an off-plane probe wave 41-43. Here, photo-elastic modulation of probe waves in planar photonic integrated circuits is used instead,' and find their proposal a good alternative to impulsive photoacoustic generation and time-resolved read out via pump & probe techniques."

"I also liked the way the author calibrated the magnitude of photo-elastic perturbations to the effective index of the waveguide. This method may prove useful in characterizing the acousto-optic conversion of optical materials, a topic of impact both for the photoacoustic and opto-electronic community."

Reply: We thank the reviewer for his/her kind words and for his/her support of our work.

Comment: *"The bibliography should be expanded in order to support some claims. Specifically: 1. On page 5, where the authors claim that "These metrics can be improved significantly in future work. The index modulation may be increased by an order of magnitude through focusing of the acoustic waves into micron-scale regions 7, such as defect cavities within Bragg gratings.", they should cite the paper from Giannetti et al. IEEE Photonics Journal 1, 21 (2009). This paper proposes a mean to localize GHz acoustic waves in a Si substrates."*

"2. On page 4, where the authors claim: "The stimulation method is scalable, in principle, to even higher frequencies, using metal patterns with shorter spatial periods 43", reference should be made to the possibility of using metallic array period down to tens of nm in order to trigger acoustic waves in the 100 GHz range. See the review from: Nardi et al., IEEE SENSORS JOURNAL, VOL. 15, No. 9 (2015)."

"3. The same reference as in point 2 should be mentioned when the authors state, on page 5, that: "The excitation of 90 GHz waveforms has been reported in bulk substrates 43."

Reply: We thank the reviewer for drawing our attention to those important references. **They are addressed in the revised manuscript.**

"For the above mentioned reasons I believe the manuscript fulfills the requirements required for publications on Nature Communications and recommend publication of the manuscript once: 1. The bibliography will be expanded. 2. The minor point hereafter reported will be taken care of."

Comment: *"Fig. 1. Periods match those OF a surface acoustic."*

Reply: The error was corrected.

Comment: "Figure 2 (a): the scale is hardly visible."

Reply: The scale bar was enlarged in all relevant panels of Figures 2, 4 and 7.

Comment: Figure (5): "The propagation losses of acoustic intensity for the fundamental surface mode are estimated as 40 ± 5 dB/mm based on the fringes visibility." Please expand in methods how the "estimate" is performed.

Reply: We thank the reviewer for this comment. An explanation for the estimate of the acoustic propagation loss coefficient was added to the Methods Section. In repeating the calculation, we have also identified and corrected an error by a factor of two.

Comment: Fig. 6.: Orange marks hardly visible. "of thee silicon-on insulator layer stack."

Reply: The typing error was corrected and the markers were replaced with clearer ones.

Avi Zadok (on behalf of all authors)

REVIEWERS' COMMENTS:

Reviewer #1 (Remarks to the Author):

In response to my comments, the authors only provide speculative answers proposing possible means to improve the excitation efficiency, acoustic loss, and modulation depth. Their answers are comprehensive and the revision helps explain the limitation as well as the prospective. Although I'm not completely satisfied, I'd like to leave the decision to the editor whether the demonstrated results have reached the level of novelty, despite the lack of technological readiness, for publication in Nature Communications.

Reviewer #2 (Remarks to the Author):

My original comments and suggestions on this paper have been convincingly addressed in the revised version of the manuscript. I think this has helped in clarifying important issues on the presented work. As mentioned in my first report on the paper, this is a very interesting piece of work worth publishing in Nature Communications. The paper is now ready to be accepted. I am confident this will attract a great deal of attention from the relevant communities.

Reviewer #3 (Remarks to the Author):

I'm fully satisfied with the changes/improvements made by the author. I therefore recommend publication of the manuscript in its present form.

Authors Reply to Reviews

The following presents our reply to the reviews of our revised manuscript: "Surface Acoustic Wave – Photonic Devices in Silicon-on-Insulator"

Response to Reviewer 1

Comment: "In response to my comments, the authors only provide speculative answers proposing possible means to improve the excitation efficiency, acoustic loss, and modulation depth. Their answers are comprehensive and the revision helps explain the limitation as well as the prospective. Although I'm not completely satisfied, I'd like to leave the decision to the editor whether the demonstrated results have reached the level of novelty, despite the lack of technological readiness, for publication in Nature Communications."

Reply: We thank the reviewer for the feedback provided, and respect his/her opinion. We detailed our roadmap for performance improvement in the current version of the manuscript, and have nothing further to add. We look to advance the technological readiness of the proposed principles in future works.

Response to Reviewer 2

Comment: "My original comments and suggestions on this paper have been convincingly addressed in the revised version of the manuscript. I think this has helped in clarifying important issues on the presented work. As mentioned in my first report on the paper, this is a very interesting piece of work worth publishing in Nature Communications. The paper is now ready to be accepted. I am confident this will attract a great deal of attention from the relevant communities."

Reply: We thank the reviewer again for his/her useful suggestions, which helped improve our paper. We are grateful for the reviewer's support of our work.

Response to Reviewer 3

Comment: "I'm fully satisfied with the changes/improvements made by the author. I therefore recommend publication of the manuscript in its present form."

Reply: We thank the reviewer for his/her positive assessment of our work.

Avi Zadok (on behalf of all authors)